# Development of a 3D simulator for training the mouse *in utero* electroporation

**Maximilian Nuber[1], Fernando Gonzalez-Uarquin[1], Meik Neufurth[2], Marc A. Brockmann[3], Jan Baumgart[1], Nadine Baumgart[1] ***

**1** Translational Animal Research Center, University Medical Center of the Johannes Gutenberg-University, Mainz, Germany, **2** Institute of Physiological Chemistry, University Medical Center of the Johannes Gutenberg-University, Mainz, Germany, **3** Department of Neuroradiology, University Medical Center of the Johannes Gutenberg-University, Mainz, Germany

* Nadine.baumgart@uni-mainz.de

**Data Availability Statement:** All relevant data files are available from the Dryad database (https://doi.org/10.5061/dryad.n8pk0p2zp).

## Abstract

In *utero* electroporation (IUE) requires high-level training in microinjection through the mouse uterine wall into the lateral ventricle of the mouse brain. Training for IUE is currently being performed in live mice as no artificial models allow simulations yet. This study aimed to develop an anatomically realistic 3D printed simulator to train IUE in mice. To this end, we created embryo models containing lateral ventricles. We coupled them to uterus models in six steps: (1) computed tomography imaging, (2) 3D model segmentation, (3) 3D model refinement, (4) mold creation to cast the actual model, (5) 3D mold printing, and (6) mold casting the molds with a mix of soft silicones to ensure the hardness and consistency of the uterus and embryo. The results showed that the simulator assembly successfully recreated the IUE. The compression test did not differ in the mechanical properties of the real embryo or in the required load for uterus displacement. Furthermore, more than 90% of the users approved the simulator as an introduction to IUE and considered that the simulator could help reduce the number of animals for training. Despite current limitations, our 3D simulator enabled a realistic experience for initial approximations to the IUE and is a real alternative for implementing the 3Rs. We are currently working on refining the model.

## Introduction

In *utero* electroporation (IUE) aims to introduce plasmid DNA into embryonic mouse brains without permanently removing embryos from the uterus [1–3]. Critical steps in IUE include the handling of the embryo in microinjection of the uterus, the genetic material in the lateral ventricle of the embryonic brain, and electroporation of the area to allow direct transfer of DNA [4, 5]. Refining these steps in the initial stages of training is crucial, as errors such as placing the load on the placenta or incorrect electroporation would burden embryos with potentially severe consequences for the procedure and the welfare of mice [6], so training is relevant to the success of IUE surgery. Here, the following question arises: How can we provide users with high-quality skills in IUE training while reducing the number of used animals? In this direction, we face educational and technological limitations; one of them is that even if we can

**Funding:** The Project was funded by the Society of Laboratory Animal Science (GV-SOLAS) J.B. and the German Research Foundation, M.A.B. INST 37/1/49-1 FUGG. The funders had no role in study design, data collection and analysis, decision to publish, or preparation of the manuscript.

**Competing interests:** The authors have declared that no competing interests exist.

create a physical model, we struggle to resemble the mechanical properties of actual organisms. Thus, improving the realism of our models could boost, at least in the first stages of training, the development of animal-free surgical procedures. Encouragingly, novel technologies are giving us new clues for designing realistic simulators while reducing the number of animals used in testing.

Developing realistic simulations for microsurgery training allows personnel to become familiar with technical and procedural knowledge, reduces the risk of suboptimal results, and improves data reproducibility. Furthermore, the implementation of animal-based simulators can help us reduce the not negligible number of animals used for education and training in Germany, which represented approximately 40.000 in 2020 [7]. Animal-based simulations promise to contribute to the 3Rs principle (replace, reduce, refine) [8], limiting the burden of laboratory animals to an unavoidable minimum while guaranteeing high standards in veterinary education and training [9–11]. Previous studies have demonstrated the success of microsurgery simulation in laboratory rodents [12, 13].

Computed tomography (CT) coupled with 3D printing creates three-dimensional imaging data and segments the final model with significant fidelity, helping to develop anatomically accurate models for high-level training or low-occurrence procedural skills [14–17]. Currently, organ 3D technology does not accurately mimic the physical properties of organ tissue, and the use of specific materials to create a realistic organ remains expensive [16, 18]; however, the development of low-cost strategies, such as silicone models cast in negative 3D printed molds, confers advanced preoperative planning and haptic simulation-based training and education for researchers and clinicians [19–23].

To our knowledge, no studies have evaluated 3D printed simulators in brain microsurgery of embryo mice. Thus, this work aimed to develop a 3D printed anatomically realistic simulator to train IUE in mice. For this purpose, we printed 3D embryo and uterus molds and filled them with silicones. The final simulator consisted of an embryo model with a lateral ventricle coupled to a uterus model. We hypothesized that the simulator would imitate the real embryo in anatomical dimensions and mechanical properties, conferring the fitness of the simulator for users to practice in an actual IUE environment.

## Materials and methods

### Mice

We performed the study according to the guidelines of the German Animal Welfare Act and the European Directive 2010/63/EU for the protection of animals used for scientific purposes. Reporting was carried out according to the ARRIVE guidelines for reporting in vivo experiments [24]. The timed pregnant C57BL/6JRj mice (specific pathogen-free [SPF], tested according to the recommendations of the Federation of Laboratory Animal Science Associations [FELASA]), were purchased on gestation day (12–14) from a registered international breeder (Janvier Labs, Le Genest-Saint-Isle, France). After arrival, the mice were housed in type II long filter top cages (Tecniplast, Buguggiate, Italy; SealSafe Plus, polyphenylsulfone, 365 mm L x 207 mm W x 140 mm H, Greenline). The mice were kept in a 12:12-h light / dark cycle (lights 06:00–18:00) in a temperature and humidity-controlled animal room (22 ± 2˚C, 55 ± 5%). Food (ssniff M-Z Extrudat, ssniff, Soest, Germany) and water were supplied ad libitum.

### Micro-computed tomography

Prior to computed tomography (CT) scanning, mothers and embryos (E15) were sacrificed with a pentobarbital overdose (Narcoren, Pentobarbital-Sodium, 16g/100ml, diluted to 400 mg/kg with 0.9% sodium chloride, Narcoren by Boeringer Ingelheim, Ingelheim, Germany).

After euthanasia, the embryos were extracted and fixed in 4% PFA (Roti Histofix, Carl Roth GmbH + Co. KG, Karlsruhe, Germany) for 48h; the PFA was exchanged after the first 24 hours. The following day, the embryos were washed in PBS (PBS tablets, 1x concentration, pH = 7.45, Gibco Thermo Fisher Scientific, Waltham, USA) overnight. To improve contrast, the embryos were dyed in a 15% Lugol solution and stored at room temperature. Subsequently, the embryos were washed and incubated in PBS overnight to be ready for CT scanning.

CT was performed using a microfocus X-ray system (Cheetah EVO YXLON GmbH, Hamburg, Germany). The embryos were placed and fixed in the sample holder with gauge bands to rotate the embryos while scanning. The voltage and current parameters were individually set per sample to achieve an optimal contrast (number of projections: 710; Scan time: 318s; Voltage: 105 kV; Current: 64 μA). After scanning, the raw data were reconstructed and exported to VG Studio Max (VG Studio Max 3.2, Volume Graphics, Heidelberg, Germany) for subsequent processing. The optimal voxel value displaying the complete surface of the embryo was selected using the isosurface renderer. With the *surface determination* in VG Studio Max, the isosurface value was chosen and subsequently rendered as an STL file.

To add the uterus to the model, a uterus containing embryos was scanned using the CT protocol described above with slight changes in the voltage and current parameters (Number of Projections: 710; Scan Time 318 s; Voltage: 105 kV, Current: 80 μA).

## Model image creation

The raw CT data were reconstructed with the "reconstruction spooler" of Yxlon. Data were adjusted, and brightness was set for optimal contrast of the object with the background using VG Studio Max. Based on the threshold value determined by a histogram representation, the Isosurface rendering was set in the 3D view window. After surface determination, the virtual STL 3D model was created.

## Model image preparation

STL files were optimized using Autodesk Meshmixer software (Autodesk, San Rafael, USA). The "shrink-smooth" function was utilized to clean tough spots. The digital model was imported as a binary STL file into Autodesk Netfabb (Autodesk, Autodesk, San Rafael, USA) using the automatic repair algorithm to readjust surfaces for anatomical correctness on the model's surface. A 20% smoothing strength was applied on the whole model. To cope with procedural flattening of the uterus (caused during scanning), copies of the uterus model were placed for a realistic alignment and merged using the "Boolean union" function.

## Lateral ventricle image creation

To create the digital model of the lateral ventricle, we implemented a volume median filter on volume (X: 3, Y: 3, Z: 3). The "pick color" function was used to determine the value of a pixel within the CT file in the VG Studio Max software. Therefore, the range of values belonging to the lateral ventricle was determined. The respective value ranges of the lateral ventricle were set. The lateral ventricle was set as the region of interest (ROI), and a virtual 3D model was created, which was finally exported as an STL file.

## Mold image creation

The embryo and lateral ventricle models were imported into Netfabb software. To fit the shape of the lateral ventricle, molds for the embryo's main body and the upper head part were created separately. For this purpose, a plane was cut through the embryo model, the lateral ventricle

model, and the surrounding box, separating all three at the same position. The lower ventricle's remaining parts were merged with the remaining boxes with the "Boolean union" function, while keeping the original parts. To create the mold for the upper head part, we subtracted the model of the upper head part from the upper part of the box. Subsequently, to create the mold for the central part of the embryo, the respective model was subtracted from the lower part remaining in the box. The mold can be opened and separated from the model after casting. To fit the size for the embryo data on embryonic day15, the mold was enlarged by 16.66%.

For the uterus model, it was necessary to create a three-part mold consisting of two outer parts containing the exterior shape of the uterus and an inner part that represents the inner space where the embryos are located. This inner part had a tab to hold inside the silicone outer mold parts; the space was limited to 0.5 mm between the inner and outer mold parts. A cube to overlap the tab with the lower end of the model and a cylinder to the upper and lower ends of the uterus model were imported from the Netfabb part library. The cylinders and the uterus model were combined with the "Boolean union" function. An additional model inside the original with the "improved hollow part" function was created, using an offset of 0.5 mm. The inner uterus model with the cylinder on each end was combined with the cube using the "Boolean union" function. The outer uterus model with both cylinders on each end and the cube was then subtracted from the outer box and the resulting mold was separated with the "plane cut" function.

## Embryo mold printing

Mold parts were exported as binary STL files and imported into preForm, slicing Formlabs (version 3.1.0, Formlabs Inc., MA, USA). Formlabs clear resin for the Form 2 stereolithography 3D printer with a resolution of 50 micrometers resolution was selected. The support structures of the embryo model were automatically generated with a diameter of 0.75 mm and a density of 1, while the uterus model was automatically generated with a diameter of 0.66 mm and a density of 0.75. Subsequently, the models were uploaded to the Form 2 Stereolithography 3D printer and printed. Post-processing of the printed models was performed following the Formlabs protocol for clear resin. The printed models were washed twice with 99% isopropanol and then post-cured with FormCure (Formlabs, Formlabs Inc., MA Somerville, USA) for 20 minutes at 60˚ C. To fit the inner mold into the outer parts, the cube had to be worked with sandpaper for an exact fit.

## Embryo model casting

Shore00-10, 20, and 30 silicones (Smooth-On, Macungie, PA, USA) were used for embryo casting. The silicones were prepared according to the manufacturer's instructions. Briefly, it consisted of two parts that must be mixed 1:1. For coloring, part B was first mixed with skin color; then part A was added, and the solution was stirred for 3 minutes. The mixture was placed in a vacuumizer for 20 min until the air was extracted. Before casting, the mold was coated with ease release (Mann Release Technologies, Macungie, PA, USA) until the silicone models were separated from the mold later. Silicones were poured into the mold for the upper head part until it was filled. The lid was then closed with the lid holding the upper part of the lateral ventricle and the parts tightly with screw clamps. To cast the main body of the embryo, the silicones were poured into separate halves of the mold before being assembled and fixing with clamps. The lid held the lower part of the lateral ventricle. The silicones were allowed to harden overnight. When releasing the embryo model from the mold, only models made of

silicone hardness Sh00-30 were left intact. Embryo models of the other harnesses (Sh00-10 and Sh00-20) stuck to the mold and were repeatedly destroyed when released.

For the uterus model, silicones were prepared as previously described, but no coloring was used in this case. Silicone was poured into both outer halves of the mold individually, the inner mold was inserted into one of them, and the remaining outer mold part was assembled. The mold was fixed with two screw clamps that allowed the silicone to harden overnight. The next day, the mold was disassembled, and the uterus model was incorporated into the inner mold part. To remove it, a subtle squeezing was applied over the inner mold. It is important not to pull on the silicone. Sh00-20 and Sh00-30 complied with the mechanical testing with similar elastic properties than the embryo. Moreover, the reason to select Sh00-30 silicone for uterus printing and implementation was the easiness to extract the model from the mold, and its handling properties to couple the model.

### IUE-model assembly

We chose both embryo and uterus models created with Sh00-30 hardness silicone to assemble the final model. Subsequently, the embryo model was lubricated with vaseline to prevent tension during insertion and pulled inside the model of the uterus. The positions of the embryo models were adjusted inside the uterus. One end of the uterus model was sutured with surgical thread and mineral oil (Carl Roth GmbH + Co. KG, Karlsruhe, Germany) was filled into the uterus, then the other end was sutured close.

### Embryo compression testing

Mice were sacrificed at E15 (as described above), and embryos were extracted. The embryos were placed under the force testing system with a 20 mm stamp and 2 N force cells. The sequence was programmed to move the stamp onto the embryo until 0,002 N was reached. The program was tared to 0 in force and distance to compress the embryo with 1 N at maximum with a movement speed of 5 mm/min. Compression distance (X-axis) and load (Y-axis) were recorded. Repetitions using the same sequence were used to test silicone embryo models.

### Uterus tensile testing

Mice were sacrificed at E15. The mice were immediately opened and cut at the upper end of the uterus and at the lower end near the cervix. The embryos were extracted through cuts to avoid damage to the uterus. The loops around the uterus were sutured with surgical-grade thread to bind to the uterus where the musculature remained constant, preserving a distance of 5 mm. Loops were used to position the uterus on hooks on the force test device. The device was tared to zero and programmed to move at a speed of 10 mm/min with a minimal force of 0,002 N. Within the measurement sequence, the force testing device moved at 10 mm/min to a maximum distance of 30 mm, where the increase in force was measured.

### Evaluation of the IUE simulator

The IUE simulation model was evaluated by presenting a practical session accompanied by a questionnaire to the IUE facility of the University Medical Center. Users were classified as beginner/intermediate (n = 3) and expert (n = 3). They were able to inspect and practice the intended steps of IUE surgery on the models. Subsequently, they received a questionnaire in which they gave their perceptions about using this model as an alternative to live-animal training. Our questionnaire was implemented to evaluate the anatomical and haptic precision of embryos and uteruses while users practice IUE surgery. The questionnaire was classified into

the following domains: User expertise (Question 1), task-based usefulness (introduction/practice to IUE surgery, questions 3 and 4), usefulness as a simulator toward the reduction/replacement of training animals (Questions 5 and 6), overall model rating (realism and assembly (Question 2), and further improvements of our simulator (Question 7). Question 1 had three possible responses (beginners, intermediate, or expert). Questions 2–6 were scaled between 1 (bad) and 10 (good), and Question 7 had a space for open responses. Responses of questions 2–6 were depicted in percentage

### Data analysis

For the Embryo Compression Testing the load [N] per displacement [mm] was compared between the embryos (n = 19) and the simulator (n = 3). Displacements of 1.5, 2 and 2.5 mm were selected for further analysis of force values, using the two-way ANOVA with Sidak's multiple comparisons. For testing the uterus displacements of 1 to 6 mm were chosen and further analyzed using 2-way ANOVA with Tukey's multiple comparison test was used.

User perception was analyzed in two ways: First, each question was individually assessed according to the user's expertise (n = 3; student's t-test with Welch's correction), to see whether training expertise might modulate their attitudes toward the simulator. Second, regardless of training experience (n = 6), the five questions were compared using a one-way ANOVA test followed by Tukey's honest significance test to evaluate user thinking toward the simulator. The bars in the graph represented the SD, and the different letters indicated significant differences. Significance was established as a P value less than 0.05.

### Results

A graphical description of the creation of embryo and uterus molds is found in **Figs 1** and **2**, respectively. The assembly of the embryo is shown in **Fig 3**.

The silicone (Sh00-30) embryo model did not show significant differences in the force required for a displacement of 1 mm (95% CI -0.3992 to 0.002982, P-value = 0.0547). Only for

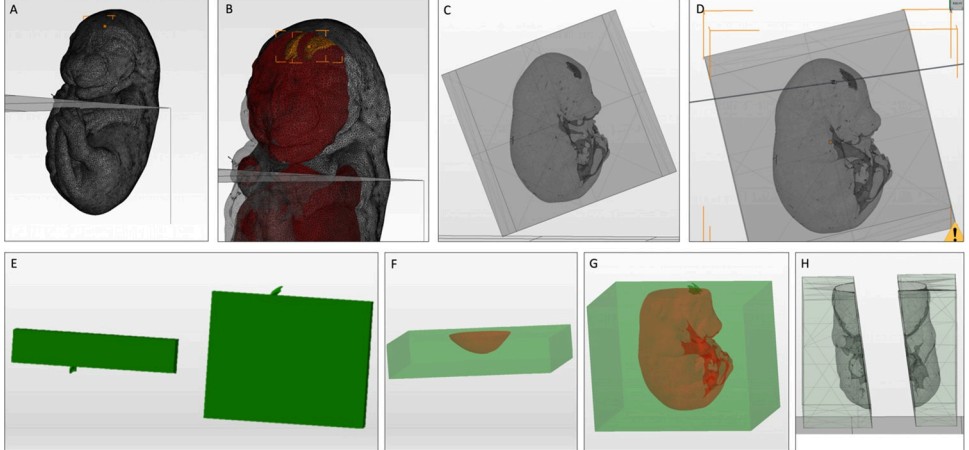

**Fig 1. Steps of mold creation from the digital embryo model.** (A) Importation of embryo model into Netfabb. (B) Importation of the posterior ventricle into Netfabb. (C) Creation of a box from the part library; it was scaled to cover the embryo model. (D) The plane cut through the box, the embryo model, and the lateral ventricle model. (E) Merging of the upper and lower parts of the lateral ventricle after the plane cut. (F) Subtraction of the head part of the embryo after the plane was cut from the surrounding box with the "Boolean difference" function. (G) Subtraction of the embryo body model after the plane is cut from w the surrounding box with the "Boolean difference" function. (H) Separation of the embryo's body mold into two halves to create the final silicon embryo.

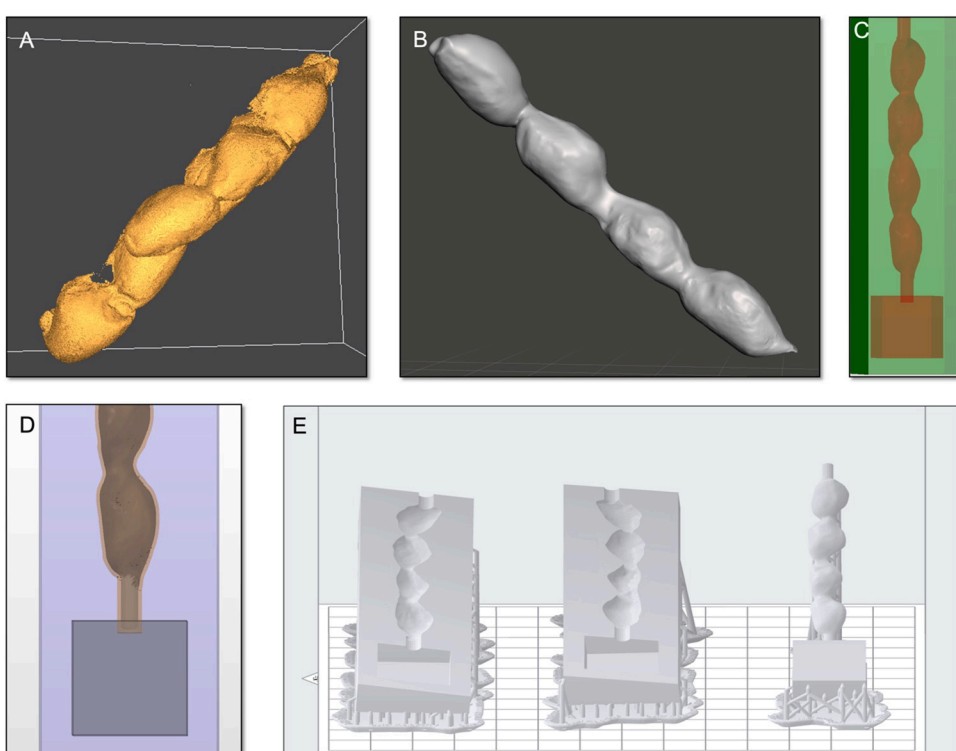

**Fig 2. Steps of mold creation from the digital uterus model.** The mold creation for the uterus model followed the same steps as for the embryo model, but used a three-part design to accommodate the anatomical tube shape of a single uterus horn. (A) and (B) Import of the uterus model into Netfabb and Meshmixer to clean up the model. (C) An additional uterus shape was created within the original, with an offset of 0.5 mm. (D)The offset uterus shape was equipped with cylinders at both ends and a cube at the lower end. (E) The original uterus model and the cube were subtracted from a surrounding box. The printed model consists of the two outer parts that outline the original uterus shape and the merged offset uterus model combined with the cylinders and the box to sit inside the two outer parts.

quite large and therefore not very realistic displacements of 2 mm (95% CI: -0.5977 to -0.1956, P-value<0.0001) and 2.5 mm (95% CI: -0.517 to -0.03578, P-value = 0.0193) significant differences were observed (**Fig 4A**). For the uterus model, we did not observe significant differences in the load required for displacements of up to 3 mm. Significant differences were observed for displacements of 4 mm between the biological samples and the simulator with Sh00-30 silicone (95% CI: -0.09717 to -0.01522, P-value = 0.0043), at 5 mm between the biosamples and simulators of both silicone hardnesses (Sh00-20: 95% CI: -0.09378 to -0.01182, P-value = 0.0079, Sh00-30: 95% CI: -0.1218 to -0.0398, P-value<0.0001), as well as for a displacement of 6 mm (Sh00-20: 95% CI: -0.112 to -0.03003, P-value = 0.0002, Sh00-30: 95% CI: -0.1502 to -0.06822, P-value<0.0001) (**Fig 4B**). For evaluation of the simulator, we only used Sh00-30 silicone. The other silicones presented a variety of problems at the moment of removing them from the mold or in the moment of the model handling (results not shown).

The embryo and uterus models fit well. The microsurgery simulation was established before user evaluation, offering a similar environment to an IUE intervention (**Fig 5**).

The evaluation of the IUE simulator based on the questionnaire (see **Table 1**) showed no differences between the level of expertise in any question (**Fig 6A–6E**). When comparing all the questions regardless of experience (**Fig 6F**), a significant decrease was observed (F-value = 5.44; P-value = 0.002). Tukey's HSD (honestly significant difference) test revealed that Question 5 (*"How would you rate the IUE simulator for replacing animals in training*

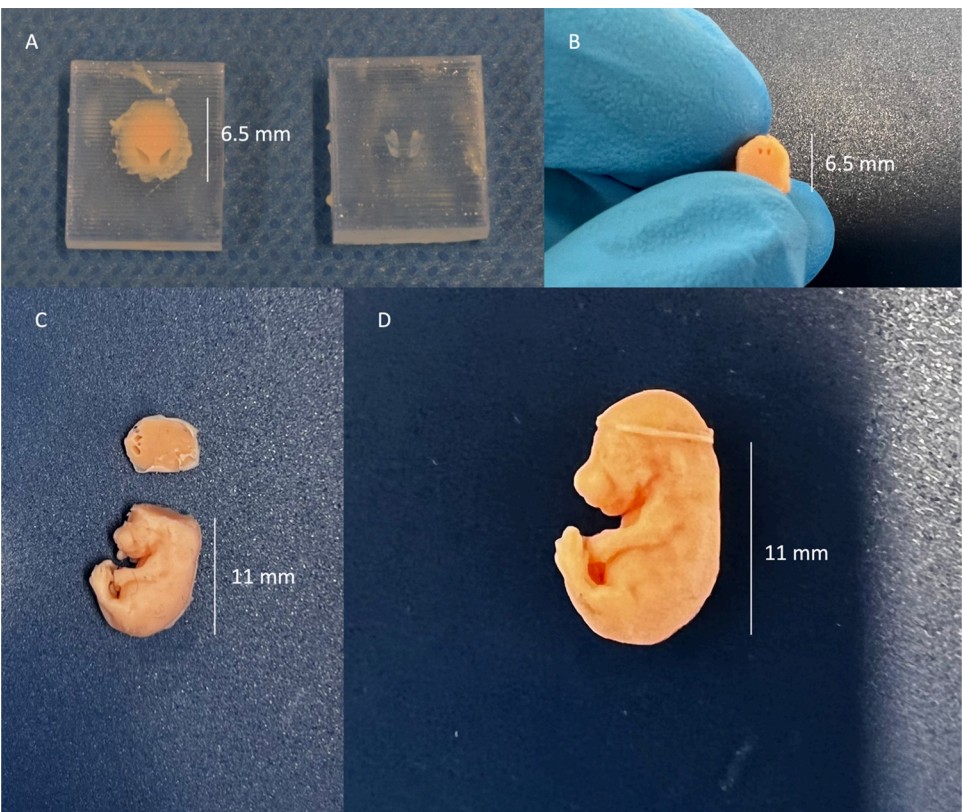

**Fig 3. Assembly of the silicone embryo model.** (A) The upper part of the head of the embryo model. The silicone was poured into the mold and hardened overnight. Due to the partial shape of the lateral ventricle on the mold cover on the right, the silicone contains two cavities representing the respective part of the lateral ventricle. (B) The model of the embryo body as seen above. Using a cover, the remaining shape of the lateral ventricle is represented as two cavities in the embryo model. (C) The upper and lower parts of the embryo model are assembled so that the two parts of the lateral ventricle meet each other. (D) The silicone embryo model was assembled with the upper and lower parts glued. The two parts of the lateral ventricle meet each other due to automatic registration in the design process, forming the complete shape of the lateral ventricle.

*completely*? ") was rated lower than all other questions. Generally, the model was rated 80% (± 6.32) (**Fig 6A**). The use of the model to introduce the IUE was remarkably well-rated (86.7% ± 10) (**Fig 6B**). The practice of specific steps was rated 72% (± 14.7) by the participants (**Fig 6C**). Using the model as a complete replacement was only rated 57% (± 20.6) (**Fig 6D**), but using it to reduce the number of animals for training was rated (88.3% ± 11.7) (**Fig 6E**).

## Discussion

We developed and provided an original and realistic 3D model for IUE training in mice. The model, composed of Sh00-30 silicone, generates a promising alternative to microsurgical training in the initial stages. This effort is particularly relevant given the worldwide efforts to implement the 3Rs principle without compromising educational and experimental reliability [8, 25] and to promote practical alternatives in education and training [12, 13].

The need to create realistic simulations (particularly complex microsurgeries such as IUE) has led to 3D printed anatomical models [26]. 3D simulators for training and skill acquisition provide users with reusability and continuous opportunity at a low cost of production. Several 3D printing strategies can recreate animal procedures accurately. Recent studies have created

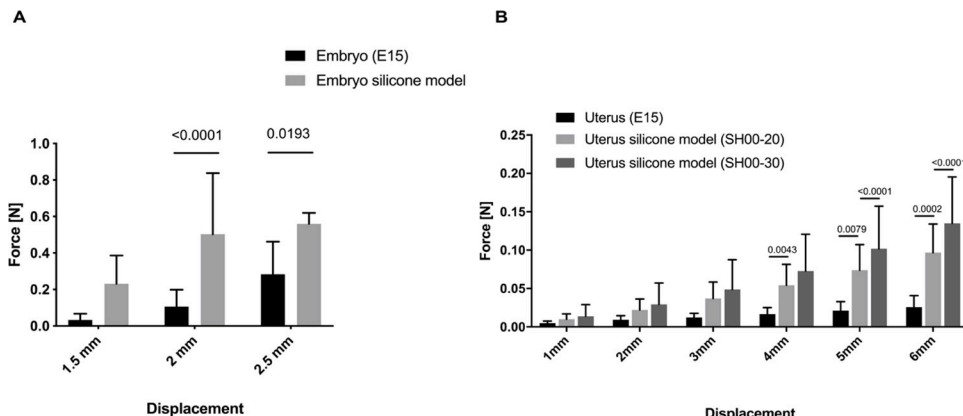

**Fig 4. Comparison of the mechanical properties from biosamples and their models.** (A) Statistical comparison between actual embryos and the simulator to evaluate the load [N] per displacement [mm] as a measure of hardness. The two-way ANOVA with Sidak multiple comparison was used to compare the statistics (Embryo's (E15): n = 19, Embryo model: n = 3). The bars in the graph represent the mean and the SD. For displacement of 1 mm: mean difference = -0.1981, 95% CI -0.3992 to 0.002982. For 1.5 mm displacement: mean difference = -0.3966, 95% CI: -0.5977 to -0.1956, P-value<0.0001. For 2 mm displacement: mean difference = -0.2764, 95% CI: -0.517 to -0.03578, P-value = 0.0193. (B) Statistical comparison between the uterus biosamples (n = 6) and the uterus silicone models (n = 6) using 2-way ANOVA with Tukey''s multiple comparison test. The test was not significant for displacements up to 3 mm. At 4 mm displacement, the comparison between E15 uteri and the Sh00-30 simulator was significantly different (uterus: mean = 0.01657, simulator: mean = 0.04885, P-value = 0.0043). At a displacement of 5 mm, the biosamples differed significantly from both artificial models (uterus(mean = 0.02107) vs. simulator Sh00-20(mean = 0.07387): P-value = 0.0002; uterus versus simulator Sh00-30(mean = 0.1018): P-value<0.0001). For the 6mm displacement, both models differed significantly from the *ex vivo* uteri as well (uterus(mean = 0.02574) vs. simulator Sh00-20 (mean = 0.09674): P-value = 0.0079; uterus versus simulator Sh00-30(mean = 0.1349): P-value<0.0001). Significance was set as P-value less than 0.05. Bars in the graph represent the SD.

successful low-cost training simulators for pediatric laparoscopic pyeloplasty and vascular anastomosis in kidney transplantation [27, 28]. The material used in 3D printing is important as well, since a prerequisite for a successful animal-based simulator is a faithful representation of the real model. In the present study, the mechanical properties presented similarities of the silicone embryo and uterus compared to the natural counterparts. Silicone with a shore hardness of SH00-20 proved to be more realistic than silicone with a shore hardness of SH00-30, so we used it for the final model. In this line, recent studies using silicone models obtained from casted molds demonstrated the benefit of haptic experience in skin and brain arteries procedures [22, 23]. Another factor to consider is that each organ differs in physical properties; therefore, we must consider specific parameters according to specific objectives [20]. Yamada et al. [29] evaluated the breaking strength and elastic modulus of a porcine heart 3D simulator and indicated that a model with numerically similar values to that obtained during actual surgery enacted a realistic experience. Similarly to our study, they achieved a realistic experience by casting models to produce a model comprising several parts. To date, no studies have investigated the physical properties of mouse embryos for training and educational processes.

Previous evidence suggested that user dissatisfaction with the available simulators usually comes with limitations in accuracy and fidelity [9, 30, 31]. In this study, the simulator provided a realistic recreation of the specific IUE setup for C57BL/6J mice [32] and had a favorable rating for providing a good introduction to IUE while reducing the number of animals for training. The size and environment provided by silicone make the surgical experience relatively realistic for planning and rehearsal [23, 27, 28, 33]. In our assessment of perception, the fact that users generally scored lower on the possibility of completely replacing the in vivo animal training with the simulator but higher on its partial implementation seemed to indicate that

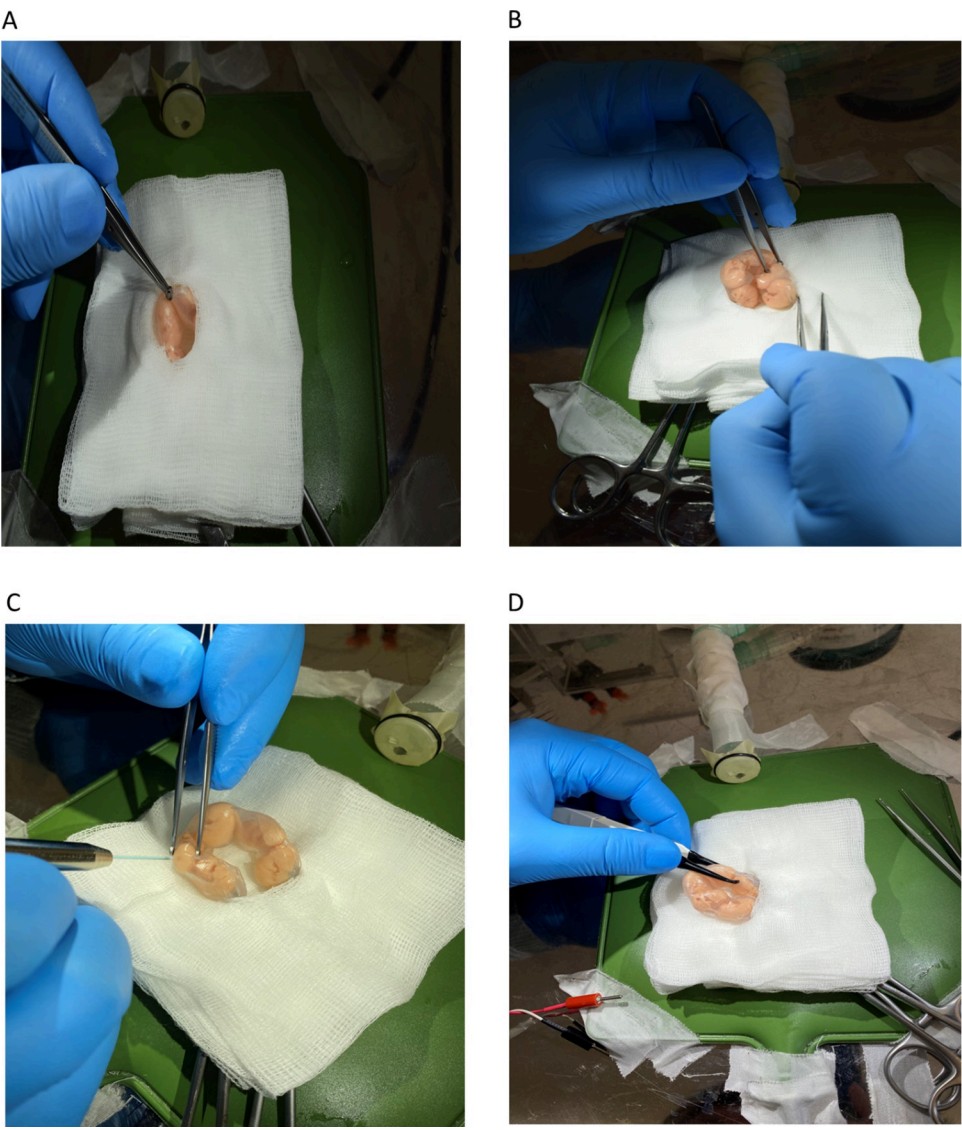

**Fig 5. Simulation of *in utero* electroporation (IUE).** (A) Opening of the mouse abdominal cavity. (B) Handling of embryos outside of the abdominal cavity. (C) Injection of the DNA-containing solution in the lateral ventricle. (D) Application of voltage through forceps-type platinum electrodes.

beginners have a slightly more positive attitude towards the simulator implementation in the first stages of training. From the 3Rs point of view, implementing our simulator in introductory training courses was effective (rather than introducing it only to experienced users). The consensus is that animals are considered the gold standard in current microsurgical training; however, combining them with simulators at the beginning of surgical training would be highly effective [30].

The development of 3D printed simulators was associated with intrinsic limitations. We observe them when recreating the intricacies of the small anatomy of mice. For example, although we designed our mold to be as large and consistent as an actual embryo, minor but significant errors could have been introduced during image segmentation to the accuracy of the STL file, affecting the model [34]. In addition, many biological components affecting the

**Table 1. Questionnaire for the *in utero* electroporation simulation model.**

| 1. What is your experience with the *in utero* electroporation? | | | | | | | | | |
|---|---|---|---|---|---|---|---|---|---|
| Beginner | | | | | | | | | |
| Advanced | | | | | | | | | |
| Expert | | | | | | | | | |
| 2. How would you rate the model overall? | | | | | | | | | |
| bad | | | | | | | | | good |
| 1 | 2 | 3 | 4 | 5 | 6 | 7 | 8 | 9 | 10 |
| 3. How would you rate the model for introducing the *in utero* electroporation? | | | | | | | | | |
| bad | | | | | | | | | good |
| 1 | 2 | 3 | 4 | 5 | 6 | 7 | 8 | 9 | 10 |
| 4. How would you rate the model for practicing the steps of the *in utero* electroporation? | | | | | | | | | |
| bad | | | | | | | | | good |
| 1 | 2 | 3 | 4 | 5 | 6 | 7 | 8 | 9 | 10 |
| 5. How would you rate the IUE simulation model for replacing animals in training? | | | | | | | | | |
| bad | | | | | | | | | good |
| 1 | 2 | 3 | 4 | 5 | 6 | 7 | 8 | 9 | 10 |
| 6. How would you rate the IUE simulation model for reducing the number of animals for training? | | | | | | | | | |
| bad | | | | | | | | | good |
| 1 | 2 | 3 | 4 | 5 | 6 | 7 | 8 | 9 | 10 |
| 7. What could be improved? | | | | | | | | | |

mechanical properties of each body tissue complicate replication. For example, much of the weight and softness of the brain consists of water, which varies according to the region of the brain, an explicit limitation for current 3D printers and materials (e.g., hydrogels and silicones) when used in small animal anatomies such as mouse brains [35]. Silicone hardness also

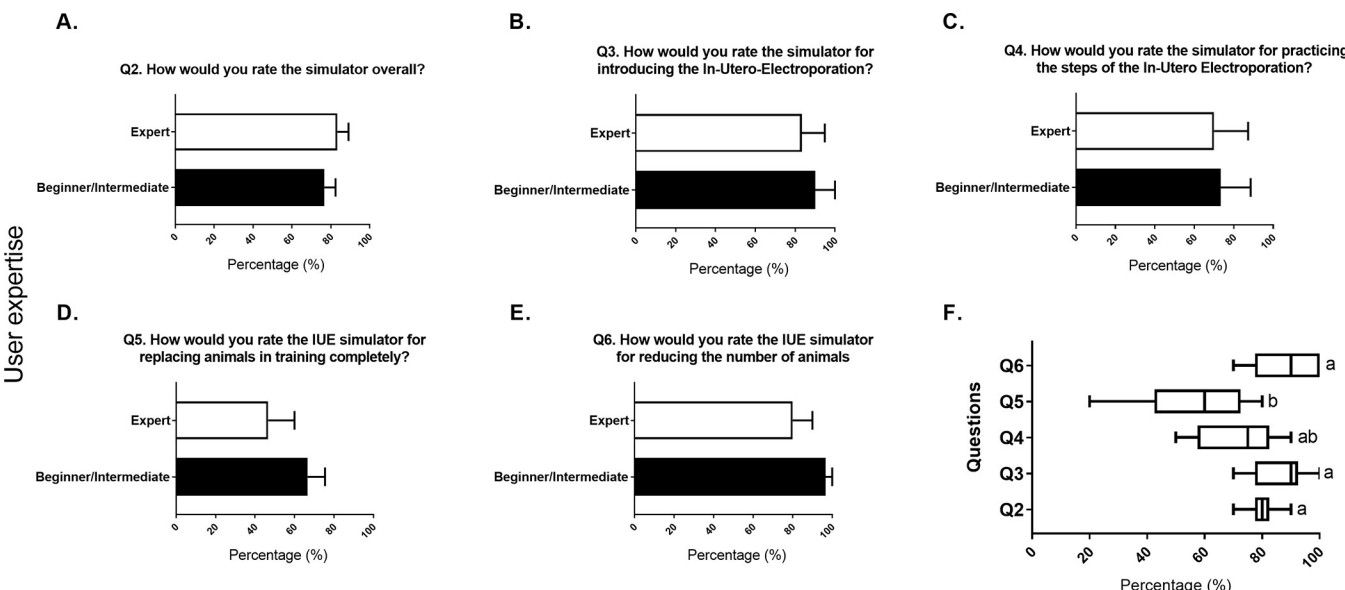

**Fig 6. Evaluation of the IUE simulator.** (A-E) The user responses to each of our questions to evaluate the IUE simulator. The X-axis indicates percentage points, while the Y-axis indicates the level of user experience. Statistical comparisons were made according to the ''user's level of expertise (Expert in white and beginner/intermediate in black). Student's t-test was used for statistical comparison (n = 3). Significance was set as P-value less than 0.05. Bars in the graph represent SD. (F) indicates the statistical comparison between the questions. The X axes indicate percentage points, whereas the Y axis indicates the respective question. The one-way ANOVA followed by Tukey's HSD (honestly significant difference) test was used for statistical comparison. Significance was set as P-value less than 0.05. Different letters indicate significant differences. Data are depicted in percentage units.

plays an important role in the moment of releasing the models from the molds and their subsequent handling (models from Sh00-10 and Sh00-20 silicones presented problems). In this respect, although the model can reproduce realism in specific properties, the whole structure may be susceptible to every assembly step. Another limitation of 3D technologies is their cost, making them inaccessible to the general public [36]. Finally, we surveyed a low number of people, however, this is based on low number of researchers trained in IUE.

## Conclusion

We developed and introduced an anatomically realistic 3D-based simulator to train IUE in mice. Our findings open possibilities for practically implementing the 3Rs principle while maintaining high-quality standards in the initial phase of training. We expect that the development of this simulator will serve as an incentive for educators and trainers to consider simulators in the initial microsurgery training stages. Additional studies are needed to assess how the overall technique can be refined to a more genuine experience. Such refinement may lie in software development to refine image segmentation and research on new printers and materials to reproduce with accuracy and reliability the brains of mice. The further implementation of simulators in the initial training stages shall assist beginners in acquiring basic knowledge about surgical techniques and a friendly attitude toward laboratory animal welfare.

## Author Contributions

**Conceptualization:** Meik Neufurth, Jan Baumgart, Nadine Baumgart.

**Formal analysis:** Maximilian Nuber, Fernando Gonzalez-Uarquin.

**Funding acquisition:** Marc A. Brockmann, Jan Baumgart.

**Investigation:** Maximilian Nuber.

**Methodology:** Meik Neufurth.

**Project administration:** Nadine Baumgart.

**Resources:** Nadine Baumgart.

**Supervision:** Nadine Baumgart.

**Visualization:** Maximilian Nuber.

**Writing – original draft:** Fernando Gonzalez-Uarquin, Nadine Baumgart.

**Writing – review & editing:** Marc A. Brockmann.

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
