## [Decision Letter · Decision Letter 0]

12 Oct 2022

PONE-D-22-25418Development of a 3D simulator for training the mouse in utero electroporation 3D simulator for in utero electroporationPLOS ONE

Dear Dr. Baumgart,

Thank you for submitting your manuscript to PLOS ONE. After careful consideration, we feel that it has merit but does not fully meet PLOS ONE’s publication criteria as it currently stands. Therefore, we invite you to submit a revised version of the manuscript that addresses the points raised during the review process.

We look forward to receiving your revised manuscript.

Kind regards,

Mohammad Mehdi Rashidi

Academic Editor

PLOS ONE

Reviewers' comments:

Reviewer's Responses to Questions

**Comments to the Author**

1. Is the manuscript technically sound, and do the data support the conclusions?

Reviewer #1: Partly

Reviewer #2: Yes

2. Has the statistical analysis been performed appropriately and rigorously? 

Reviewer #1: No

Reviewer #2: I Don't Know

3. Have the authors made all data underlying the findings in their manuscript fully available?

Reviewer #1: Yes

Reviewer #2: Yes

4. Is the manuscript presented in an intelligible fashion and written in standard English?

Reviewer #1: Yes

Reviewer #2: Yes

5. Review Comments to the Author

Reviewer #1: Thank you for this opportunity to review this study. I understand the aim of this study is to evaluate the anatomical reality of 3D printed simulator for in utero electroporation (IUE) in mice. The topic is important and timely; however, several points are required to revise.

1) In introduction, the statement of background issues is vague. For instance, if this research purpose is to reduce the number of mice to sacrifice, it should be described in the first paragraph of introduction, then evaluate how much declined using the simulator. While the main purpose is to evaluate the level of anatomical accuracy of 3D printed simulator, as this study compared the embryo compression testing, current technological/educational boundaries need to be stated in the first paragraph in detail. Depending on the background issue to solve, the purpose of this study might differ.

2) Materials and method are well described; however, some statements require more explanation. For instance, page 5, line 2 from the bottom, “The resulting model might be altered by specialized programs, as discussed in the following sections”, might be confusing for readers.

3) In the paragraph of embryo model casting, the reason why three different silicones were used for embryo model and only one silicone Sh00-30 was used for the uterus model. The explanation might be too in detail, e.g., “Silicones were poured into the mold for the upper head part until it was filled. The lid was then closed …, etc.

4) There is a severe lack of explanation for the evaluation of the IUE simulator. Table 1 might be the demographic characteristic of the participants (beginners/intermediate and experts). Clear statement of questionnaire is required.

5) In data analysis, the number of simulators and the number of the participants are confusing. The 19 embryos and the 3 simulators were compared in Figure 4, however, it is unknown that the participants evaluated the three simulator to answer the questionnaire.

6) The first paragraph of Discussion should be the summary of the obtained results, for instance which silicone was the best for simulator. An additional table will be beneficial for Figure 4, so other researchers can avoid to conduct the same study.

7) Numbers and units were missing in Figure 6.

8) Conclusion requires to answer the aim of study.

I hope these comments are helpful.

Thank you for this opportunity.

Reviewer #2: The manuscript showed that it is possible to develop skill simulators with more accessible costs, using simpler technologies, such as the 3D printer Form 2. The study manages to circumvent the printing limitation of its equipment by printing negative molds that would later be filled with silicone at different Shore scales reproducing the reality of the simulators that were required. Therefore, it comes close to the result of sophisticated and high technology printers that use multiple raw materials, different mixtures and have greater value.

The authors concluded that the low-cost simulator was well accepted and that it can be used in the training and qualification of the electroporation technique in utero. Strengthening and meeting the 3R principles (reduce, replace and refine) in the use of live animals in scientific experiments.

6. PLOS authors have the option to publish the peer review history of their article (what does this mean?). If published, this will include your full peer review and any attached files.

Reviewer #1: **Yes: **Noriyo Colley

Reviewer #2: No

---

## [Author Response · Author response to Decision Letter 0]

2 Nov 2022

Reviewer #1:

1) In introduction, the statement of background issues is vague. For instance, if this research purpose is to reduce the number of mice to sacrifice, it should be described in the first paragraph of introduction, then evaluate how much declined using the simulator. While the main purpose is to evaluate the level of anatomical accuracy of 3D printed simulator, as this study compared the embryo compression testing, current technological/educational boundaries need to be stated in the first paragraph in detail. Depending on the background issue to solve, the purpose of this study might differ.

We thank you very much for your valuable comment. In this study, we purposed to create a realistic model for the training of the IUE electroporation. The implementation of the simulator in the practice can contribute to reducing the number of animals used for IUE training, but this must be confirmed in further studies (thank you for a such relevant idea). Following your observation, we included more detail about technological and educational limitations in the first part of the introduction:

Line 59 to 67: “…, so training is relevant to the success of IUE surgery. Here, the following question arises: How can we provide users with high-quality skills in IUE training while reducing the use of animals? In this direction, we face educational and technological limitations; one of them is that even if we can create a simulator, we struggle to resemble the mechanical properties of actual embryos. Thus, in the first step, improving the realism of our models could boost, at least in the first stages of training, the development of animal-free surgical procedures. Encouragingly, novel technologies are giving us new clues for designing realistic simulators while reducing the number of animals used in testing.”

2) Materials and method are well described; however, some statements require more explanation. For instance, page 5, line 2 from the bottom, “The resulting model might be altered by specialized programs, as discussed in the following sections”, might be confusing for readers.

We thank you for your suggestion. We removed the sentence from the main text. 

3) In the paragraph of embryo model casting, the reason why three different silicones were used for embryo model and only one silicone Sh00-30 was used for the uterus model. The explanation might be too in detail, e.g., “Silicones were poured into the mold for the upper head part until it was filled. The lid was then closed …, etc.

Thank you for your important remark. Unfortunately, we were negligent in mentioning that embryo models made from silicones with harnesses of Sh00-10 and Sh00-20 were not feasible to implement in further evaluation. The principal problem of Sh00-10 and Sh00-20 was to release them out of the mold. We definitely found the embryo model with Sh00-30 silicone to be more than satisfactory. 

Likewise, for the uterus silicone model, we used only Sh00-30 models to create practical models for participants of the questionnaire. We chose this model based on our experience in handling the models. Please find the information within the manuscript, as follows:

Line 199 to 201: “…When releasing the embryo model from the mold, only models made of silicone hardness Sh00-30 were left intact. Embryo models of the other harnesses (Sh00-10 and Sh00-20) stuck to the mold and were repeatedly destroyed when released.”

Line 208 to 211: “… Sh00-20 and Sh00-30 complied with the mechanical testing with similar elastic properties than the embryo. Moreover, the reason to select Sh00-30 silicone for uterus printing and implementation was the easiness to extract the model from the mold, and its handling properties to couple the model.”

Line 213 to 214: “… We chose both embryo and uterus models created with Sh00-30 hardness silicone to assemble the final model.”

Line 306 to 308: “… For evaluation of the simulator, we only used Sh00-30 silicone. The other silicones presented a variety of problems at the moment of removing them from the mold or in the moment of the model handling (results not shown).” 

Lines 464 to 468: “Silicone hardness also plays an important role in the moment of releasing the models from the molds and their subsequent handling (models from Sh00-10 and Sh00-20 silicones presented problems). In this respect, although the model can reproduce realism in specific properties, the whole structure may be susceptible to every assembly step.”

4) There is a severe lack of explanation for the evaluation of the IUE simulator. Table 1 might be the demographic characteristic of the participants (beginners/intermediate and experts). Clear statement of questionnaire is required.

We utterly agree with your comment. In consequence, we included a more detailed description of the evaluation of the simulator. A new statement of the questionnaire was added, as follows:

Line 241 to 249: “… Our questionnaire was implemented to evaluate the anatomical and haptic precision of embryos and uteruses while users practice IUE surgery. The questionnaire was classified into the following domains: User expertise (Question 1), task-based usefulness (introduction/practice to IUE surgery, questions 3 and 4), usefulness as a simulator toward the reduction/replacement of training animals (Questions 5 and 6), overall model rating (realism and assembly (Question 2), and further improvements of our simulator (Question 7). Question 1 had three possible responses (beginners, intermediate, or expert). Questions 2-6 were scaled between 1 (bad) and 10 (good), and Question 7 had a space for open responses. Responses of questions 2-6 were depicted in percentage.”

5) In data analysis, the number of simulators and the number of the participants are confusing. The 19 embryos and the 3 simulators were compared in Figure 4, however, it is unknown that the participants evaluated the three simulator to answer the questionnaire.

Thank you very much to permit us to give clarification about this point. As described in our response to your third question, embryo models made from Sh00-10 and Sh00-20 silicones were not viable for use in the practice, due to practical issues in releasing them from the mold. Therefore, we only used embryo and uterus silicone models of Sh00-30 hardness. Please see the amendments made to the manuscript in lines 199–201, 208–211, 213–214, 306–308, and 464 – 468.

6) The first paragraph of Discussion should be the summary of the obtained results, for instance which silicone was the best for simulator. An additional table will be beneficial for Figure 4, so other researchers can avoid to conduct the same study.

We thank you for the essential note. As we mentioned, we used only one Shore hardness of Sh00-30 for the final model. Thank you, because due to your useful and helpful comments, we could make clear in the manuscript the fact that there is only one version of the finalized model. We regret not having data from the models made from the other silicones (beyond the shown in Figure 4) but we also see the potential future work for refining the processes in terms of model elaboration, realism, and mechanical properties. We highlight this information in the manuscript:

Line 199 to 201: “…When releasing the embryo model from the mold, only models made of silicone hardness Sh00-30 were left intact. Embryo models of the other harnesses (Sh00-10 and Sh00-20) stuck to the mold and were repeatedly destroyed when released.”

Line 208 to 211: “… Sh00-20 and Sh00-30 complied with the mechanical testing with similar elastic properties than the embryo. Moreover, the reason to select Sh00-30 silicone for uterus printing and implementation was the easiness to extract the model from the mold, and its handling properties to couple the model.”

Line 213 to 214: “… We chose both embryo and uterus models created with Sh00-30 hardness silicone to assemble the final model.”

Line 306 to 308: “… For evaluation of the simulator, we only used Sh00-30 silicone. The other silicones presented a variety of problems at the moment of removing them from the mold or in the moment of the model handling (results not shown).” 

Line 414 to 416: “… We developed provide an original and realistic 3D model for IUE training in mice. The model, composed of Sh00-30 silicone, generates a promising alternative to microsurgical training in the initial stages.” 

Lines 464 to 468: “Silicone hardness also plays an important role in the moment of releasing the models from the molds and their subsequent handling (models from Sh00-10 and Sh00-20 silicones presented problems). In this respect, although the model can reproduce realism in specific properties, the whole structure may be susceptible to every assembly step.”

7) Numbers and units were missing in Figure 6.

Thank you for such a constructive comment. It makes us realize other corrections in terms of numbering and description. We amended the manuscript, as follows:

Lines 335 to 344: “The evaluation of the IUE simulator based on the questionnaire (see Table 1) showed no differences between the level of expertise in any question (Fig 6A-E). When comparing all the questions regardless of experience (Fig 6F), a significant decrease was observed (F-value=5.44; P-value=0.002). Tukey's HSD (honestly significant difference) test revealed that Question 5 ("How would you rate the IUE simulator for replacing animals in training completely? “) was rated lower than all other questions. Generally, the model was rated 80 % (± 6.32) (Fig 6A). The use of the model to introduce the IUE was remarkably well-rated (86.7 % ± 10) (Fig 6B). The practice of specific steps was rated 72 % (± 14.7) by the participants (Fig 6C). Using the model as a complete replacement was only rated 57 % (± 20.6) (Fig 6D), but using it to reduce the number of animals for training was rated (88.3 % ± 11.7) (Fig 6E).”

Line 393: “...Data are depicted in percentage units.”

Figure 6 was replaced.

8) Conclusion requires to answer the aim of study.

We apologize for not making this clear. We added more information to our conclusion, making sure we answered our research question and aim.

Line 473 to 475: “We developed and introduced an anatomically realistic 3D-based simulator to train IUE in mice. Our findings open possibilities for practically implementing the 3Rs principle while maintaining high-quality standards in the initial phase of training...”

I hope these comments are helpful.

Thank you for this opportunity.

We thank you for your time!

Reviewer #2: The manuscript showed that it is possible to develop skill simulators with more accessible costs, using simpler technologies, such as the 3D printer Form 2. The study manages to circumvent the printing limitation of its equipment by printing negative molds that would later be filled with silicone at different Shore scales reproducing the reality of the simulators that were required. Therefore, it comes close to the result of sophisticated and high technology printers that use multiple raw materials, different mixtures and have greater value.

The authors concluded that the low-cost simulator was well accepted and that it can be used in the training and qualification of the electroporation technique in utero. Strengthening and meeting the 3R principles (reduce, replace and refine) in the use of live animals in scientific experiments.

We thank you very much for taking the time for assessing our manuscript. Your comments reflect our purpose of implementing and developing this work in pro of animal welfare.

---

## [Editor Report · Decision Letter 1]

29 Nov 2022

Development of a 3D simulator for training the mouse in utero electroporation 3D simulator for in utero electroporation

PONE-D-22-25418R1

Dear Dr. Baumgart,

We’re pleased to inform you that your manuscript has been judged scientifically suitable for publication and will be formally accepted for publication once it meets all outstanding technical requirements.

Kind regards,

Mohammad Mehdi Rashidi

Academic Editor

PLOS ONE
---

## [Editor Report · Acceptance letter]

5 Dec 2022

PONE-D-22-25418R1 

Development of a 3D simulator for training the mouse *in utero* electroporation 

Dear Dr. Baumgart:

I'm pleased to inform you that your manuscript has been deemed suitable for publication in PLOS ONE. Congratulations! Your manuscript is now with our production department. 

Kind regards, 

on behalf of

Professor Mohammad Mehdi Rashidi 

Academic Editor

PLOS ONE